# Molecular Targets, Pathways, and Therapeutic Implications for Hepatocellular Carcinoma

**DOI:** 10.3390/ijms21155232

**Published:** 2020-07-23

**Authors:** Jun Gong, Jeremy Chuang, May Cho, Kyra Toomey, Andrew Hendifar, Daneng Li

**Affiliations:** 1Department of Gastrointestinal Malignancies, Cedars-Sinai Samuel Oschin Comprehensive Cancer Institute, Los Angeles, CA 90048, USA; Jun.Gong@cshs.org; 2Department of Medical Oncology, City of Hope Comprehensive Cancer Center, Duarte, CA 91010, USA; jechuang@coh.org; 3Department of Internal Medicine, UC Davis Comprehensive Cancer Center, Sacramento, CA 95817, USA; maycho@ucdavis.edu (M.C.); katoomey@ucdavis.edu (K.T.); 4Department of Medicine, Cedars-Sinai Medical Center, Los Angeles, CA 90048, USA; andrew.hendifar@cshs.org

**Keywords:** unresectable, hepatocellular, gastrointestinal, cancer, biomarkers

## Abstract

Hepatocellular carcinoma (HCC) represents one of the leading causes of cancer mortality worldwide. While significant advances have been made for the treatment of advanced hepatocellular carcinoma in the past few years, the prognosis remains poor and effective biomarkers to guide selection of therapies remain noticeably absent. However, several targeted therapies have been approved in the past few years that have improved the outlook for this disease. In this review, we will highlight the recent therapies approved for the treatment of advanced HCC and discuss promising therapeutic options, targets, and pathways for drug development and consideration for future clinical trials.

## 1. Introduction

Primary liver cancer represents the fifth most common cancer but the second leading cause of cancer death globally [1]. Hepatocellular carcinoma (HCC) is the most common primary liver cancer, accounting for approximately 90% of cases [2]. The most common risk factors for HCC are well-described and include chronic viral hepatitis B and C, alcohol consumption, and aflatoxin exposure [3]. Globally, approximately 54% of HCC cases are attributable to hepatitis B virus (HBV) infection, with a greater predominance in Africa and East Asia, while 31% can be attributed to hepatitis C virus (HCV) infection. Cirrhosis is also an important risk factor for HCC and can be caused by chronic HBV and HCV infection, alcoholism, nonalcoholic fatty liver disease (NAFLD), and rare causes including inherited metabolic diseases such as hemochromatosis or alpha-1-antitrypsin deficiency.

Treatment and outcomes of HCC vary by stage of disease and classification system [3]. Using the Barcelona-Clínic Liver Cancer (BCLC) classification system, median overall survival (OS) can exceed 60 months for very early stage (stage 0) and early stage (stage A) tumors whereby standard treatments include surgical resection, liver transplantation, and radiofrequency ablation (RFA). For intermediate stage (BCLC B) HCC, median OS approximates 20 months, whereby locoregional therapies including transarterial chemoembolization (TACE) are standard options. Prognosis is poor in advanced stage (BCLC C) HCC where median OS approximates 11 months and systemic therapies are predominately recommended. In BCLC stage D (end-stage) HCC, prognosis is exceedingly poor with a median OS < 3 months and, as a result treatment, is not often considered.

In advanced (unresectable or metastatic) HCC where prognosis remains dismal and systemic therapies represent the cornerstone for treatment, efforts have been focused on the development of more effective systemic therapies to improve outcomes in this population. In contrast to other cancers where molecular classifications have guided targeted therapies (e.g., HER2 in breast cancer, EGFR in non-small cell lung cancer, or RAS/BRAF in colorectal cancer), there exists no molecular subclass of HCC that has an approved biomarker-driven therapy.

However, targeted therapies in the form of inhibitors of vascular endothelial growth factor (VEGF) and its receptor (VEGFR), MET, RAF, platelet-derived growth factor receptors (PDGFR), programmed cell death protein 1 (PD-1), and cytotoxic T-lymphocyte-associated protein 4 (CTLA-4) are now established as standard systemic therapy options in advanced HCC. The nature of these approvals provide insight that HCC is indeed responsive to select targeted therapies. The lack of a definitive biomarker to guide such therapies, however, underscores the need for further understanding of molecular targets and pathways that can have therapeutic implications in HCC. This is especially relevant in the setting of advanced HCC, where improvements in therapeutic outcomes are desperately needed.

In this review, we first highlight the systemic therapies that have been approved and the molecular targets of these drugs in advanced HCC. We then discuss recent seminal findings characterizing somatic genomic alterations that may be of potential therapeutic importance and look forward to future implications in advanced HCC.

## 2. Current Molecular Targets of Approved Systemic Agents

A number of agents are currently used as the standard of care for first-line, second-line, and subsequent line therapies for advanced HCC (Table 1). Sorafenib is a tyrosine kinase inhibitor that was U.S. Food and Drug Administration (FDA) approved in 2007 for use as first-line therapy for advanced HCC [4]. The drug functions by inhibiting tumor cell proliferation and angiogenesis by targeting the vascular endothelial growth factor receptors (VEGFRs), platelet-derived growth factor receptor β (PDGFR-β), and the serum-threonine kinases Raf-1 and B-Raf. The approval was based on the results of the multicenter, randomized phase III SHARP trial that randomly assigned 602 patients with treatment-naïve advanced HCC to receive 400 mg of either sorafenib or placebo twice daily. The median overall survival (OS) was 10.7 months in patients treated with sorafenib vs. 7.9 months for placebo (hazard ratio (HR) 0.69; 95% confidence interval (CI) 0.55–0.87, *p* < 0.001). There was no significant difference in time to symptomatic progression (4.1 months vs. 4.9 months *p* = 0.7). In total, seven patients (2%) were reported to have achieved a partial response compared to 2 patients (1%) in the placebo arm. Sorafenib was well-tolerated with a rate of discontinuation similar to the control arm (38% vs. 37%). The most common grade ≥3 treatment-related adverse events include diarrhea and hand-foot syndrome.

Lenvatinib is a multi-kinase inhibitor that inhibits VEGFR1–3 as well as fibroblast growth factor receptors (FGFR) 1–4 along with PDGFR-α, c-KIT, and rearranged during transfection (RET) protooncogene. FDA approval in the first-line setting was granted in 2018 after results were reported from the phase III REFLECT non-inferiority clinical trial which enrolled 954 treatment-naïve patients with unresectable or metastatic HCC who received either 8 mg or 12 mg of lenvatinib daily vs. 400 mg of sorafenib twice daily [5]. In this study, the median overall survival was 13.6 vs. 12.3 months (HR 0.92; 95%CI 12.1–14.9) demonstrating non-inferiority of lenvatinib to sorafenib. The most common adverse events included hypertension, diarrhea, and decreased appetite.

Regorafenib is an oral multi-kinase inhibitor with antiangiogenic activity against VEGFR2 that was FDA approved in 2017 as second-line therapy for advanced HCC based on results reported from the international, multicenter randomized phase III RESORCE clinical trial [6]. The 573 patients who progressed on sorafenib with Child-Pugh A liver function were enrolled and randomized 2:1 to receive either 160 mg of regorafenib or placebo daily. Overall survival in the experimental cohort was 10.6 months vs. 7.8 months (0.63; 95%CI 0.50–0.79; one-sided *p* < 0.0001). The most common grade ≥3 adverse events include hypertension, hand-foot syndrome, fatigue, and diarrhea.

Nivolumab, a PD-1 inhibitor, received accelerated FDA approval in 2017 for patients with advanced HCC who progressed on or were intolerant/refused first-line sorafenib [7]. Approval was granted following the results of phase I/II dose escalation and expansion trial CHECKMATE-040 which enrolled 262 patients with advanced HCC and Child-Pugh A cirrhosis who progressed on or were intolerant/refused to take sorafenib. Participants of the clinical trial were administered 3 mg/kg of nivolumab by intravenous infusion every 2 weeks in the dose-expansion phase to patients in 4 cohorts: HBV infected, sorafenib untreated or intolerant without viral hepatitis, sorafenib progressor without viral hepatitis, and HCV infected. The primary endpoints were safety and tolerability with 12 patients (25%) demonstrating grade ≥3 adverse events. The objective response rate (ORR) was 20% (95%CI 15–26) in patients treated with 3 mg/kg of nivolumab. However, confirmatory phase III study checkmate 459 enrolled 743 treatment-naïve patients with advanced HCC and observed no significant difference in overall survival between nivolumab and sorafenib with an OS of 16.4 months in the experimental arm vs. 14.7 months in the control group (HR 0.85; 95%CI 0.72–1.02, *p* = 0.0752) [8].

Pembrolizumab is another PD-1 inhibitor that was also granted accelerated FDA approval as second-line therapy for advanced HCC based off results reported from the single-arm phase II KEYNOTE-224 clinical trial [9]. One-hundred and four patients with HCC previously treated with sorafenib with Child-Pugh A were given 200 mg of pembrolizumab as an intravenous infusion every three weeks until disease progression or unacceptable toxicity. The primary endpoint of objective response rate was reported to be 18/104 patients (17%) with 1 (1%) complete response and 17 (16%) partial responses according to RECIST 1.1 criteria. Common grade ≥3 adverse events include elevated Aspartate aminotransferase (AST), Alanine transaminase (ALT), and hyperbilirubinemia. The confirmatory phase III keynote 240 clinical trial did not meet statistical significance for the overall survival of pembrolizumab versus placebo, raising the question if single agent PD-1 inhibition is sufficient as a target in HCC [10]. The median OS was 13.9 months in the experimental arm vs. 10.6 in the placebo arm (HR 0.781; 95%CI 0.611–0.998; *p* = 0.0238).

Cabozantinib is a receptor tyrosine kinase inhibitor that targets MET, VEGFR, RET, GAS6 receptor (AXL), KIT, and FMS-like tyrosine kinase-3 (FLT3). The drug was FDA approved in 2019 for second-line therapy after results of the randomized phase III CELESTIAL clinical trial. A total of 707 patients with advanced HCC and Child-Pugh A liver disease who progressed on sorafenib [11] were randomized in a 2:1 ratio to receive 60 mg daily of either cabozantinib or placebo. Median OS was 10.2 vs. 8.0 months (HR 0.76; 95%CI 0.63–0.92; *p* = 0.005) and progression-free survival (PFS) was 5.2 vs. 1.9 months (HR 0.44; 95%CI 0.36–0.52; *p* < 0.001) in favor of cabozantinib. Grade ≥3 adverse events were more common in the experimental arm (68% vs. 36%). The most common adverse events were palmar-plantar erythrodysesthesia, hypertension, and increased AST.

Ramucirumab is a VEGFR2 antagonist that was FDA approved in 2019 as second-line therapy in patients with advanced HCC previously treated with sorafenib with an alpha fetoprotein (AFP) ≥ 400 ng/mL [12]. Approval was granted based on the phase III REACH-2 randomized, double-blinded clinical trial that enrolled 292 patients with advanced Child-Pugh A HCC to receive either 8 mg/kg of ramucirumab intravenously every 2 weeks or placebo. This was the first positive phase III clinical trial utilizing a biomarker selected population. The primary endpoint of overall survival was 8.5 months in the cohort randomized to ramucirumab vs. 7.3 months in the placebo cohort (HR 0.71; 95%CI 0.531–0.949 *p* = 0.0199). The most common grade ≥3 adverse events include hypertension, hyponatremia, and elevated AST.

Nivolumab and ipilimumab are PD-1 and CTLA-4 inhibitors, respectively, that was recently granted accelerated FDA approval on 10 March 2020 for second-line therapy in advanced HCC previously treated with sorafenib. Approval was granted based on the results of phase I/II CHECKMATE-040 that enrolled 148 patients and was randomized into three cohorts: A: 1 mg/kg of nivolumab and 3 mg/kg of ipilimumab every 3 weeks (*n =* 50); B: 3 mg/kg of nivolumab and 1 mg/kg of ipilimumab every 3 weeks followed by 240 mg of nivolumab every 2 weeks (*n =* 49); or C: 3 mg/kg of nivolumab every 2 weeks and 1 mg/kg of ipilimumab every 6 weeks (*n =* 49). The primary endpoint was safety and tolerability. Overall, 14% of patients discontinued treatment secondary to toxicity and 37% of patients experienced grade ≥3 adverse events with the most common adverse events including rash and pruritus. The 24-month overall survival rate was 40%, and the overall response rate was observed in 46/148 patients (31%).

Combination therapies have also shown promise in advanced HCC with atezolizumab and bevacizumab, demonstrating improved overall survival and progression-free survival compared to sorafenib [13]. In the phase III IMbrave150 clinical trial of 501 patients with unresectable HCC, the OS at 12 months was 67.2% in the combination therapy cohort vs. 54.6% with sorafenib. The HR for death was 0.58; 95%CI 0.42–0.79; *p* < 0.001. Median PFS was 6.8 months vs. 4.3 months (HR 0.59; 95%CI 0.47–0.76; *p* < 0.001). This combination was subsequently FDA approved on 29 May 2020 for the first-line treatment of advanced HCC [13].

## 3. Landmark Comprehensive Genomic Analyses

The advent of massive parallel sequencing with high-throughput functionality has allowed several groups to perform more in-depth genomic profiling of HCC in an effort to identify prognostic biomarkers and molecular alterations of potential therapeutic significance [15,16]. The most common somatic gene mutations, as prior described, in HCC coalesce into signaling pathways that have varied, and putative roles in the development of HCC (Figure 1) and many of these are in pathways considered druggable [17]. Recent studies have identified aberrations within the Telomerase Reverse Transcriptase (TERT) promoter region as central to hepatocarcinogenesis with up to 70% of tumors identified with this mutation [18]. Other key mutations identified include TP53 and aberrations within the Wingless-related integration site (WNT)/β-catenin cell signaling pathway with up to 45% of patients harboring one of these mutations [19]. In addition, 25% of tumors involve either IDH1 missense mutations, MET amplifications, FGF19, or TSC1/2.

### 3.1. Telomerase Reverse Transcriptase (TERT)

Telomerases are enzymes that prevent the degradation of the ends of chromosome following repeated cycles of DNA replication. Telomerase reverse transcriptase (TERT) is a subunit within the telomerase enzyme that specifically catalyzes the addition of nucleotides to the ends of telomeres within a chromosome, thereby prolonging the life of senescent cells that otherwise would undergo apoptosis [20]. Telomerase dysregulation specifically with mutations in TERT has been postulated to play a major role in oncogenesis, particularly in hepatocarcinogenesis. Specifically, studies within The Cancer Genome Atlas (TCGA) has identified that patients with high levels of C15orf55 and C7orf43, positive regulators of TERT expression, are involved in hepatocarcinogenesis and that, compared to patients with low levels of these regulators, these patients have poorer overall survival [21].

### 3.2. Cell Cycle Signaling

Several mutations within tumor suppressors including TP53, RB1, CDKN2A, CCNE1, and FBXW7 have also been implicated in hepatocarcinogenesis. TP53 is one of the most frequently associated aberrations in human cancer with inactivation mutations identified in 33% of patients with HCC [19]. Low p53 activity has been associated with tumors with more advanced pathological grade and increased risk of tumor recurrence [22]. CDKN2A is a gene that codes for the tumor suppressor proteins p16^INK4A^ and p14^ARF^ and was also identified to be frequently mutated within HCC. Collectively, 72% of tumors analyzed were identified to have aberrations in either one of these signaling pathways. Aberrations within CDK1 and CDKN1A, important modulators of the cell cycle, may predict resistance to treatment with sorafenib [23]; deficiency of FBXW7, a suppressor, has been associated with increased radio sensitivity in HCC [24].

### 3.3. Chromatin Remodeling

Chromatin is a complex that primarily comprises of histone proteins that covalently bind to DNA to form a nucleosome and enable the packing of DNA in a compact unit [25]. Alterations to the histone-DNA complex results in changes in gene expression [26]. Several genes have been implicated to play a role in the chromatin modification including ARID, BAP1, Histone-lysine *N*-methyltransferase 2D (KMT2D), and cAMP response element-binding protein (CREBBP) with 50% of HCC patients found to have an alteration in one of these pathways. ARID1A plays an important role in the regulation of transcription by altering the structure of chromatin and gene expression. Specifically, ARID1A has been observed to be associated with increased tumor burden and therefore may portend response to immune checkpoint inhibitors [27]. In HCC, ARID1A deficiency has been observed to be associated with HCC carcinogenesis by activating the angiopoietin-2 angiogenesis pathway; sorafenib has been shown to suppress this specific pathway and ARID1A deficiency may confer sensitivity to treatment with sorafenib [28].

### 3.4. WNT/β-Catenin Cell Signaling Pathway

Dysregulation of the WNT/β-catenin cell signaling pathway has also been implicated in carcinogenesis. There are a variety of modulators that have been identified as part of this pathway including CTTNB1, APC, and AXIN1. Missense mutations within CTTNB1 are the most commonly reported aberrancy within this pathway [26]. Dysregulation within this pathway has been observed in up to 66% of patients with HCC in genome studies [18]. In addition, recent studies have found that aberrations within this pathway may also be associated with worsened progression-free survival and overall survival, particularly in patients treated with sorafenib [19]. One of the downstream effectors of this pathway is the gene *MYC* that is a proto-oncogene that encodes for transcription factors. Aberration within the expression of *MYC* has been observed to portend resistance to sorafenib [23].

### 3.5. Phosphatidylinositol-3-Kinase (PI3K)/AKT/Mammalian Target of Rapamycin (mTOR)

The phosphatidylinositol-3-kinase (PI3K)/AKT/mammalian target of rapamycin (mTOR) is a cell signaling pathway critical to cellular growth, proliferation, and angiogenesis [29]. A number of different growth factors including MET, VEGF-α, and FGFR are known to activate this signaling pathway while Phosphatase and tensin homolog (PTEN) and TSC1/2 are known to counteract these modulators [30]. Approximately 45% of patients with HCC have aberrations within the P13K/AKT/mTOR pathway, with the most common mutations identified within the TSC1/2 complex [18]. In addition, aberrations within this pathway have been observed to be associated with decreased progression-free survival and overall survival in patients treated with checkpoint inhibitors [19].

### 3.6. RAS/Mitogen-Activated Protein Kinase (MAPK) Pathway

The RAS/MAPK pathways is a signal transduction pathway comprising of serine/threonine kinases that play an important role in cellular proliferation, differentiation, and angiogenesis that has been implicated in tumorigenesis. In HCC, activation of this pathway has been observed in over 50% of tumors and is associated with a poor prognosis with decreased survival [31]. Sorafenib, a multikinase inhibitor targeting downstream kinases from RAS including Raf-1 and B-raf within this signaling pathway was the first approved systemic treatment for advanced HCC to demonstrate an improvement in overall survival [4].

### 3.7. Nuclear Factor-Like 2 (NFE2L2)/KEAP1 Pathway

Nuclear factor-like 2 (NFE2L2) is a transcriptional factor that regulates the processes involved with metabolism and antioxidation [32]. It is regulated by KEAP1; mutations involving this pathway in either NFE2L2 or KEAP1 leading to oxidative stress are a known factor in hepatocarcinogenesis [26].

### 3.8. Additional Biomarkers of Interest

Chronic hepatitis B and C infections are well-established risk factors for the development of HCC. Within the TCGA dataset, 44/196 (22.4%) patients demonstrated clinical and/or molecular evidence of HBV infection while 35/196 (17.9%) patients exhibited evidence suggestive of HCV infection. HBV-positive HCCs were associated with mutations involving TP53 and less so with TERT compared to patients with HBV-negative HCC [22,33]. In contrast, HCV-positive HCC tends to harbor CDKN2A silencing mutations and mutations within the TERT promoter region.

Checkpoint inhibitors that target cytotoxic T-lymphocyte antigen 4 (CTLA4) and programmed cell death 1 (PD1) have changed the treatment paradigm across a variety of different cancers over the past few years. The TCGA identified 43/196 (22%) of cases of HCC that demonstrated moderate to high levels of lymphocyte infiltration [22]. Six clusters of tumor samples demonstrated high expression of sixty-six cell-surface immune markers in patients with HCC within the TCGA dataset including CTLA4, PD-1, and PD-L1. There was no association with overall survival or an association with HBV/HCV infection.

## 4. Novel Druggable Targets

From these landmark genomic analyses, alterations were found in genes considered to be targetable but have yet to have approved agents in HCC [18,22].

### 4.1. Fibroblast Growth Factors (FGFs)

Fibroblast growth factors (FGFs) are growth factors that signal via tyrosine kinase fibroblast growth factor receptors (FGFRs) which are involved in the regulation of cellular differentiation, proliferation, and tissue repair [34]. Dysregulation through this pathway is recognized to be a contributor to hepatocarcinogenesis [35]. Specifically, dysfunction within fibroblast growth factor receptor 4 (FGFR4) has been implicated as a contributor to increased cellular proliferation and increased proclivity towards metastasis [36]. Targeting of this pathway has shown promise in HCC with the recent approval of lenvatinib which is a known FGFR inhibitor [37].

Several trials exploring the role of novel agents targeting FGFR have been disappointing. Orantinib, a multi-kinase inhibitor, was evaluated in a randomized phase III clinical trial vs. placebo in combination with transcatheter arterial chemoembolism (TACE) in 889 Asian patients with advanced HCC with Child-Pugh ≤6 [38]. Patients were randomized to receive 200 mg of orantinib twice daily or placebo. There was no improvement in overall survival in the experimental cohort compared to the control arm (HR 1.09; 95%CI 0.878–1.352 *p* = 0.435). The most common grade ≥3 adverse events included elevated AST, ALT, and hypertension. The randomized phase III BRISK-FL evaluated the role of brivanib, a dual inhibitor of VEGFR and FGFR, in 1150 treatment-naive patients with advanced HCC [39,40]. Patients were randomized 1:1 to 400 mg of sorafenib twice daily orally (*n =* 578) or 800 mg of brivanib once daily orally (*n =* 577). The primary end point of OS non-inferiority was not met (HR 1.06; 95% CI, 0.93 to 1.22). Median OS was 9.9 months in the sorafenib cohort vs. 9.5 months for brivanib.

Despite these disappointing results, a number of experimental drugs are currently being studied in phase I clinical trials that target the FGFR4 pathway including H3B-6527 (NCT02834780), U3-1784 (NCT0260350), BLU-554 (NCT02508467), and INCB62079 (NCT03144661) [41,42,43].

### 4.2. c-KIT

c-KIT is a proto-oncogene that expresses the tyrosine kinase protein CD117. In a recent molecular study, 6/258 (2.3%) of patients with HCCs expressed positive immunohistochemical staining for c-KIT [44]. Given the promise of imatinib in the role of gastrointestinal stromal tumors, a phase II clinical trial studied the role of imatinib in patients with c-KIT positive advanced HCC but found little to no efficacy for the drug [45]. Pazopanib has also been studied in a phase I clinical trial in 28 Asian patients with advanced HCC with 19/28 (73%), demonstrating partial response or stable disease [46]. Other drugs that have been studied include sunitinib in a randomized phase III trial versus sorafenib in 1074 patients with advanced HCC [47]. However, in this study, sunitinib had significantly inferior OS compared to sorafenib (7.9 versus 10.2 months HR 1.30 *p* = 0.0014) while experiencing more frequent and severe toxicities. While there have been limited successes with sorafenib, lenvatinib, and regorafenib in inhibiting c-KIT, additional studies are warranted.

### 4.3. JAK-STAT Cell Signaling Pathway

The JAK-STAT cell signaling pathway plays a significant role in cell proliferation, death, and aberrations within this pathway and has been implicated in carcinogenesis. Sorafenib is a multi-kinase inhibitor that has activity across a number of targets including this pathway [48]. A phase I clinical trial evaluating the role of OPB-111077, a novel STAT3 inhibitor, has shown promise in 33 patients with advanced HCC who were previously treated with sorafenib. The therapy was well tolerated. However, no patients have demonstrated achieving either a partial or complete response. Nevertheless, additional studies are warranted to identify whether this may be a potential therapeutic target for patients with HCC.

### 4.4. Epidermal Growth Factor Receptors (Egfrs)

A number of studies have evaluated targeting the epidermal growth factor receptor (EGFR), a transmembrane protein with downstream effects involving cellular proliferation and growth. A meta-analysis of eight phase II clinical trials including 342 patients with advanced HCC previously treated with sorafenib found that the combination therapy was an effective option for treating sorafenib-refractory HCC [49]. The pooled ORR was 12% while the 6- and 12-month OSs were 77.8% (95% CI: 71.3–84.2%) and 44.9% (95% CI: 36.8–53.0%). Other phase I/II studies that evaluated EGFR inhibitors including cetuximab, vandetanib, ganetespib, and lapatinib showed limited clinical benefit in patients with advanced HCC [50,51,52,53].

### 4.5. c-MET

c-MET, also known as hepatic growth factor receptor (HGFR), is a protein encoded by the MET gene that plays a significant role in embryonic development, organogenesis, and wound healing. Recently, cabozantinib, a tyrosine kinase inhibitor, was FDA approved as second-line therapy for patients with HCC previously treated with sorafenib. Several studies have evaluated the role of novel MET inhibitors to treat patients with advanced HCC. A phase I/II trial evaluated the role of foretinib as first-line therapy in patients with advanced HCC [54]. The study observed an ORR of 22.9% with a median OS of 15.7 months. IL-6 and -8 were identified as potential predictive biomarkers to therapy. Single-agent capmatinib demonstrated promise in a phase II study in a subset of 38 patients with elevated MET expression in advanced HCC with an overall response rate of 30% including 1 patient with a durable complete response and 2 patients achieving partial responses [55]. Another novel agent, tepotinib, was evaluated in 49 patients with advanced MET^+^ HCC previously treated with sorafenib in a single-arm phase II study [56]. Thirty-one of 49 patients (63%) were progression free at 12 weeks, and 4/49 patients (8.2%) had an objective response with 3 partial responses. The median OS was 5.6 months. A randomized phase III evaluated the role of tivantinib in patients previously treated with sorafenib; however, the study drug did not demonstrate an improvement in overall survival compared with placebo [57]. Combination therapy with MET inhibitors is also being evaluated as options as well. A phase I/II clinical trial found that the combination of golvantinib and sorafenib was able to achieve partial responses in 2/12 patients (17%) in a study of 13 patients with refractory advanced HCC [58].

### 4.6. Isocitrate Dehydrogenase (IDH)

Interestingly, in the TCGA dataset, four patients with HCC were identified to harbor the IDH1/2 mutations, oncogenic mutations involving isocitrate dehydrogenase, which plays an important role in producing nicotinamide adenine dinucleotide phosphate (NADPH) [22]. This is significant in light of the recent results from the randomized, phase III ClarIDHy trial evaluating the role of ivosidenib vs. placebo in 186 patients with IDH1-positive unresectable or metastatic cholangiocarcinoma. The authors reported a significant improvement in progression-free survival of 2.7 vs. 1.4 months (HR 0.37 *p* < 0.001). The response rate was 2.4% but with stable disease reported in 50.8% of patients. Median OS was 10.8 vs. 9.7 months with ivosidenib vs. placebo, respectively.

### 4.7. ATM

ATM serine/threonine kinase is a key component of the DNA repair pathway that is specifically activated when DNA double-strand breaks occur. Preclinical data suggests that agents that target HCC cells may provide promise of antitumor activity. In one study, the ATM inhibitor KU-55933 was able to enhance the antitumor effect of sorafenib by inhibiting cell proliferation and by inducing tumor cell apoptosis [59]. Another preclinical study found that palbociclib may potentially be another agent of interest given it was found to enhance radiosensivity both in vivo and in vitro by inhibiting the ATM pathway in HCC [60].

### 4.8. P13K/AKT/mTOR Signaling Pathway

As discussed previously, the P13K/AKT/mTOR pathway plays a significant role in cellular proliferation and aberrations in this pathway have been implicated in carcinogenesis. SF1126, an inhibitor of the PI3K/mTOR pathway, has been evaluated in both preclinical and clinical models as a promising agent that is well-tolerated with antitumor effects in patients with advanced malignancies; however, additional studies will be needed to evaluate its clinical impact [61,62]. Another novel agent, MK-2206, an AKT inhibitor, was given in combination with the mTOR kinase inhibitor AZD8055 and MEK inhibitor AZD6244, and preclinical models showed that these agents work synergistically to inhibit cell proliferation in HCC [63]. An extensive number of trials investigating mTOR inhibition in HCC have yet to establish an effective mTOR inhibitor. However, there may be potential for combination therapy involving inhibition of the PIK3CA/mTOR pathway moving forward [64].

## 5. Future Directions

Despite recent large-scale comprehensive genomic profiling efforts that have better characterized the molecular signatures of HCC and uncovered novel targets of potential therapeutic relevance, the occurrence of actionable mutations in HCC are relatively rare. Although clinical trials investigating the efficacy of systemic agents targeting these mutations are ongoing, many have been negative or have produced mixed results. An established biomarker for current FDA-approved agents in advanced HCC also does not exist. Instead of focusing on single somatic gene alterations in HCC for development of prognostic and predictive biomarkers, several groups have attempted broad molecular classification of HCC into distinct subclasses to more comprehensively and accurately identify subgroups for relevance to prognosis and therapy design.

### 5.1. Transcriptomics

A robust transcriptome analysis identified six subgroups of HCC that were associated with distinctive clinical and genetic markers [65]. HCC groups 1–3 were associated with a higher rate of chromosomal instability compared to groups (G) 5–6 and commonly more often associated with HBV infection. G2–3 were observed to demonstrate higher rates of TP53 mutations, while G5–6 were associated with hypermethylation and were associated with E-cadherin (CDH1) and mutations of CTNNB1. The results of this analysis demonstrate the genetic diversity and markers within HCC that may herald future therapeutic interventions, particularly within specific cell signaling pathways including WNT and AKT pathways. A different transcriptome metanalysis analyzed 603 patients with HCC [66]. Three subclasses denoted S1–3 were identified. S1 was defined by mutation within the WNT signaling pathway and transforming growth factor (TGF)-β. S2 was distinguished by activation of the MYC and AKT signaling pathway along with elevation in AFP and EpCAM. AKT and P13K inhibitors may be potential therapeutic targets within this subclass. Finally, S3 was associated with hepatocyte differentiation, preserved p53 function, and small tumor size indicating less aggressive tumors. Further study is still required to determine whether this classification may help to guide the design of future clinical trials and to influence the decision of choosing therapeutic options.

### 5.2. Genomics

Several genome-wide studies of HCC have also led to different classification schemas for HCC. A genomic-based study from 307 patients with HCC sought to identify and validate gene-expression patterns that were associated with survival and recurrence [67]. The first conclusion in the study was the lack of a tumor genetic profile that was associated with survival; however, the authors hypothesized that genetic profile of the surrounding liver tissue may harbor a signature that may portend the emergence of a second primary tumor rather than a true recurrence of the primary tumor. Using gene-expression profiles, the study was able to identify 113 genes associated with a good-prognosis signature including those associated with normal liver function including complement proteins, alcohol dehydrogenase, etc. In contrast, the poor-prognosis subset included 73 genes that were associated with inflammation, signaling of tumor necrosis factor α, and interferon signaling. The expression patterns for both good and poor prognosis were validated with observed differences in overall survival. A second analysis within this dataset also yielded factors associated with recurrence. Factors that were associated with early recurrence, defined as within 2 years after therapy, included tumor multi-nodularity, presence of microvascular invasion, and elevated serum alpha-fetoprotein.

### 5.3. Integrated Transcriptomics, Genomics, and Proteomics

The TCGA was able to identify three subtypes (iC1–iC3) based on molecular profiling with important clinical correlation [22]. iC1 tumors were associated with patients with younger age, female gender, Asian ethnicity, and normal body weight and were defined molecularly by low rates of CTNNB1 mutations. iC2 cohort was distinguished by lower grade tumors and lower rates of microvascular invasion. iC3 was characterized by increased DNA hypomethylation. Overall, iC1 was associated with better prognosis while iC2 and iC3 were associated with poor prognosis. Other classification proposals include a Japanese genomic study of 183 HCC tumor specimens identifying three major subtypes distinguished by AFP levels, vascular invasion, chromosomal instability, and mutations within TP53 [68]. There have also been attempts to characterize HCC by CTNNB1, proliferation, inferno-related, and polysomy 7 [69].

Another classification proposed includes stratification by epithelial cell adhesion molecule (EpCAM) and alpha-fetoprotein (AFP), resulting in four subgroups [70]. These subgroups were found to have distinct genetic profiles, specifically EpCAM and AFP positive tumors were associated with activation of the WNT/β-catenin pathway while mature hepatocyte-specific genes were associated with EpCAM negative AFP-positive tumors. Additional analyses observed that AFP-positive HCC correlated with poor prognosis, EpCAM and AFP-negative denoted intermediate prognosis, and EpCAM-positive and AFP-negative HCC correlated with good prognosis. A major limitation of this study is that these cases represent resectable diseases and that it is unclear whether these results may be reflected in unresectable or more advanced diseases. Taken together, these findings suggest that different therapeutic approaches may be required to address this heterogeneous entity.

Furthermore, it is becoming increasingly recognized that comprehensive molecular characterization of HCC would need to entail analysis beyond genomic profiling. Here, beyond tissue genomic analyses, proteomics, metabolomics, and epigenomics are future avenues of exploration to identify novel prognostic and predictive biomarkers in HCC [15,16]. Within genomics, analysis of copy number variations (CNVs), which has been reviewed extensively elsewhere, will also be useful in HCC given that amplifications in other tumor types have translated to effective biomarker-driven treatments [15]. Liquid biopsies (circulating tumor DNA and circulating tumor cells) are also exciting and noninvasive strategies to profile HCC tumors [71].

### 5.4. Molecular Heterogeneity

Ongoing molecular profiling of HCC will also need to take into account issues of intertumoral heterogeneity. Of significant importance is the contribution that hepatitis B and C infection plays in hepatocarcinogenesis and of how different studies have demonstrated that each virus has a unique mutation profile [72,73]. HBV-related HCC is associated with hypermethylation of CCND1, TP53 mutations, TERT, and MLL4. These mutations are related to the integration of viral DNA into the host genome. In contrast, HCV is an RNA virus that does not integrate into the host genome and is associated with mutations within TERT, CKDN2A, and ARID2. Other etiologies of HCC include aflatoxin and alcohol which each have unique mutational profiles [74]. Aflatoxin-induced HCC was associated with TP53 while alcohol-induced HCC was found to be associated with alterations in TERT, CDKN2A, and ARID1A.

Intratumoral heterogeneity is the recognition that, even within one tumor, there can exist multiple subpopulations with different molecular and genetic makeup. This was first described in an early study that identified that up to half the patients with HCC had at least two or more subpopulations even within a single tumor [75]. A separate study of 23 patients with HCC identified 87% patients having tumors demonstrating intratumoral heterogeneity [76]. In addition, the larger the tumor or more advanced the disease, the more genetic heterogeneity available. An additional study found that, in 10 HCC patients, all tumor samples were found to have tumor heterogeneity, which is consistent with the previous studies that intratumoral heterogeneity is common and likely underappreciated [77]. Analysis of the genetic and molecular profiles of the various clones suggests possible branched evolution leading to intratumoral heterogeneity. In addition to subpopulations, differences at the single-cell level are important to recognize. A study of 25 cell-lines within one HCC tumor demonstrated that there were two distinct subgroups distinguished by different molecular patterns [78]. In all, these findings of intratumoral heterogeneity suggest that a single sample from a tumor may not adequately characterized the depth of the complexity of the lesions that we may be attempting to understand or treat. They may also help to explain a reason for resistance to therapy and early recurrence.

## 6. Conclusions

Despite recent advances and a plethora of newly approved therapies, hepatocellular carcinoma’s treatment options remain limited with poor prognosis. In addition, many clinical trials of targeted agents over the past few years have yielded mixed results, with few if any actionable targets or biomarkers to guide treatment selection. Comprehensive genomic studies have identified hepatocellular carcinomas with distinct molecular and genomic markers that may lead to promising new therapeutic options and may identify novel genes and pathways for drug development. Several classification schemas have arisen over the past decade that demonstrate the heterogeneous and complex nature of this disease. However, further study is required to understand their prognostic and clinical significance in making treatment decisions for our patients.

## Figures and Tables

**Figure 1 ijms-21-05232-f001:**
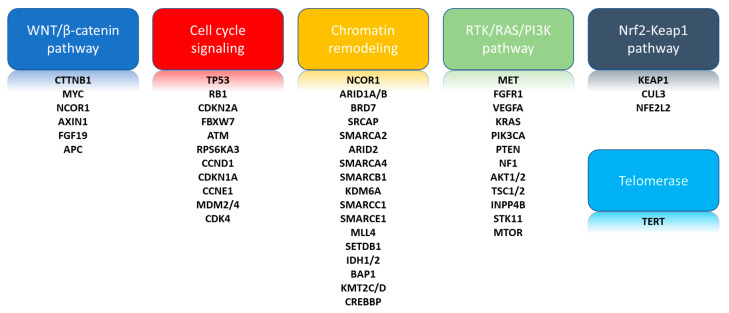
Major gene mutations and pathways altered in hepatocellular carcinoma.

**Table 1 ijms-21-05232-t001:** Approved systemic therapies in advanced hepatocellular carcinoma.

Study	Setting	Treatment Arms	Molecular Targets	Results	Citation
SHARP, phase III	First-line	Sorafenib (*n =* 299) vs. placebo (*n =* 303)	Sorafenib: VEGFRs, PDGFR-β, and Raf-1, and B-Raf	OS: 10.7 vs. 7.9 months (HR 0.69; 95%CI 0.55– 0.87, *p* < 0.001) Time to symptomatic progression: 4.1 vs. 4.9 months (*p* = 0.7)	[4]
REFLECT, Phase II	First-line	Lenvatinib (*n =* 478) or sorafenib (*n =* 476)	Lenvatinib: VEGFR1–3, FGFR 1–4, PDGFR-α, c-KIT, and RET	OS: 13.6 vs. 12.3 months HR 0.92; 95%CI 12.1–14.9) PFS: 7.3 vs. 3.6 (HR 0.64; 95%CI 0.55–0.75 *p* < 0.001) ORR: 41% vs. 12%	[5]
RESORCE, phase III	Second-line	Regorafenib (*n =* 374) vs. placebo (*n =* 193)	Regorafenib: VEGFR2 inhibitor	OS: 10.6 vs. 7.8 months (HR 0.63; 95%CI 0.50–0.79 *p* < 0.0001)	[6]
CHECKMATE-040, phase I/II	Second-line	Nivolumab (*n =* 262)	Nivolumab: PD-1	Safety and tolerability with 12 patients (25%) demonstrating grade ≥3 adverse events ORR: 20% (95%CI 15–26)	[7]
KEYNOTE-224, phase II	Second-line	Pembrolizumab (*n =* 104)	Pembrolizumab: PD-1	ORR 18/104 patients (17%) with 1 (1%) complete response and 17 (16%) partial responses	[9]
CELESTIAL, phase III	Second-line	Cabozantinib (*n =* 467) vs. placebo (*n =* 237)	Cabozantinib: MET, VEGFR, RET, GAS6 receptor (AXL), KIT, and FMS-like tyrosine kinase-3 (FLT3)	OS: 10.2 vs. 8.0 months (HR 0.76; 95%CI 0.63–0.92 *p* = 0.005) PFS: 5.2 vs. 1.9 months (HR 0.44; 95%CI 0.36–0.52 *p* < 0.001) ORR 4% vs <1% (*p* = 0.009)	[11]
REACH-2, phase III	Second-line	Ramucirumab (*n =* 197) vs. placebo (*n =* 95)	Ramucirumab: VEGFR2 antagonist	OS: 8.5 vs. 7.3 months (HR 0.71; 95%CI 0.531–0.949 *p* = 0.0199) PFS: 2.8 vs. 1.6 months (HR 0.452; 95%CI 0.339–0.603) *p* < 0.0001)	[12]
Phase I/II	Second-line	Nivolumab (N) and ipilimumab (I). Cohort A: (N 1mg/kg + I 3 mg/kg (*n =* 50); cohort B: N 3 mg/kg + I 1 mg/kg (*n =* 49); cohort C: N 3 mg/kg + I 1 mg/kg (*n =* 49)	Nivolumab: PD-1; Ipilimumab: CTLA-4	Grade ≥ 3 adverse event 37% ORR: A: 16/32 (16%); B: 15/31 (15%); C: 15/31 (15%) 24-month OS: A: 61%; B: 30%; C: 42%	[14]

OS, overall survival; PFS, progression-free survival; HR, hazard ratio; CI, confidence interval; ORR, overall response rate; VEGFR, vascular endothelial growth factor receptor; FGFR, fibroblast growth factor receptor; PDGFR, platelet-derived growth factor receptor; PD-1, programmed cell death 1; CTLA, cytotoxic T-lymphocyte-associated; mg, milligram; kg, kilogram.

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
