# Peer review of "Molecular Targets, Pathways, and Therapeutic Implications for Hepatocellular Carcinoma"

_ijms, 2020, doi:10.3390/ijms21155232_

Round 1
Reviewer 1 Report
In the present manuscript, Gong et al. provide a comprehensive overview of the molecular alterations occurring in human hepatocellular carcinoma (HCC) as well as on the actual (approved) and possible therapeutic strategies against this deadly disease. In addition, the authors discuss the molecular pathways that could be relevant for drug development and consideration for future clinical trials.
The review article by Gong et al. is interesting, detailed, and summarizes the current knowledge on the molecular pathogenesis and related therapies for HCC. Tables and figures are properly designed and significantly help the reader to better understand the topic. Some issue should be addressed to further improve the quality of the present review article:
1. Besides Table 1, a figure summarizing the pathways and related approved targeted therapies would be highly helpful for the readers.
2. The authors should discuss the issue of undruggable targets (c-Myc, ARID1A, etc.) and the possible strategies to overcome this problem in HCC.
3. Although the Ras genes are rarely mutated in this tumor type, the Ras/MAPK is often activated in HCC and should be discussed.
4. Minor polishing of the text is suggested.
Author Response
- We have added to table 1 the molecular targets of all the approved therapeutic agents that concisely summarizes the different pathways involved. (Line 161 within the new table is an additional column)
- We have expanded our discussion of a number of undruggable targets under the sections of cell cycle signaling, chromatin remodeling, and WNT/B-catenin cell signaling pathway to specifically discuss the roles of C-myc, ARID1A, and FBXW7 as predictive biomarkers of response to sorafenib. (Lines 194-196, 202-208, and 216-218)
- A new section has been added to discuss the important role of RAS/MAPK pathway in HCC. (Lines 228-235)
- With the help of reviewer 2, we were able to make changes that enhances the readability of this text.
Reviewer 2 Report
This review paper is clearly written while some small modification is required before publishing.
1, In line 301, there is a statement that “C-MET also known as hepatic growth factor (HGF)”. Actually, C-MET is hepatocyte growth factor receptor (HGFR).
2, In line 320, the author may describe what is IDH first.
3, Some small mistakes need to be modified such as line 291: Epidermal growth factors (EGFRs).
Author Response
1. Changes made.
2. IDH is clarified.
3. Changes made.